

# Evaluating the spatial heterogeneity of soil loss tolerance and its effects
# on erosion risk in the carbonate areas of South China
Yue Li[1,2],   Xiaoyong Bai[※1,2,4], Shijie Wang[1,2], Luoyi Qin[1,2] ,
Yichao Tian[1,3], Guangjie Luo[1,3]
1. State Key Laboratory of Environmental Geochemistry, Institute of Geochemistry, Chinese Academy of Sciences,
Guiyang, Guizhou, 550002, PR China
2. Puding Comprehensive Karst Research and Experimental Station, Institute of Geochemistry, CAS and Science and
Technology Department of Guizhou Province, Puding, Guizhou, 562100, PR China
3. Graduate School of Chinese Academy of Sciences, Beijing 100029, PR China
4. Institute of Mountain Hazards and Environment, Chinese Academy of Sciences,Chengdu, Sichuan, 610041, PR
China
*Correspondence to:* Bai xiaoyong (baixiaoyong@126.com)
**Abstract.** Soil loss tolerance ($T$ value) is the ultimate criterion to determine the necessity of erosion control
measures and ecological restoration strategy. However, the validity of this criterion in subtropical karst regions is
strongly disputed. In this study, $T$ value is computed based on soil formation rate by using a digital distribution map
of carbonate rock assemblage types. Results indicated spatial heterogeneity and diversity in such values; moreover,
a minimum of three criteria should be considered instead of only one criterion when investigating the carbonate
areas of South China given that the "one region, one $T$ value" concept may not apply to this region. $T$ value is
proportionate to the amount of argillaceous material in formations that determine surface soil thickness in
homogenous carbonate rock areas; such values are 20 and 50 t/(km²·a) in carbonate rock intercalated with clastic
rock areas and 100t/(km²·a) in carbonate/clastic rock alternation areas. These three areas are each extremely,
severely, and moderately sensitive to soil erosion. This erosion is extreme in karst rocky desertification (KRD) land
and reflects the degree of erosion risk. Thus, the relationship between $T$ value and erosion risk is determined with
KRD as a parameter. The existence of KRD land is unrelated to $T$ value, although this parameter indicates erosion
sensitivity. In fact, erosion risk is strongly dependent on the relationship between real soil loss (RL) and $T$ value
rather than on either erosion intensity or the $T$ value itself. If RL >> $T$, then erosion risk is high despite a low RL.
Conversely, if $T$ >> RL, the soil is safe although RL is high. Overall, these findings may clarify $T$ value
heterogeneity and its effect on erosion risk in a karst eco-environment; hence, innovative technological assessment
solutions need not be invented.
**1 Introduction**

The fragile ecological environment of karst areas is closely related to surface soil (Bülent Turgut, Merve Ateş.





2016; Nigussie Haregeweyn et al., 2017). However, these factors are less associated with the total lack of inherent
soil in such areas (Zhongwu Li et al., 2017; İlknur Gümüş; Xu et al., 2013). Soil is continuously distributed through
erosion, and rocky desertification landscapes are frequently generated (Tegegne Molla and Biniam Sisheber 2016).
Determining soil loss tolerance ($T$ value) is one of the most important criteria in controlling erosion and restoration
ecosystems; therefore, this factor must be measured scientifically and rationally. $T$ is expressed in terms of annual
soil loss (t/km$^2$·a) and reflects the maximum level of soil erosion that can occur while allowing the land to sustain
an indefinite, economic level of crop productivity (Wischmeier and Smith 1965, 1978). This value is an important
criterion in determining the potential erosion risk of a particular soil and often serves as the ultimate erosion control
criterion to preserve long-term soil productivity (Duan et al., 2012). Thus, a scientifically determined $T$ value is
among the most significant aspects in the planning of soil erosion control on agricultural lands and on other types of
lands (Liu et al., 2003). The concept of this value was first proposed in the United States in 1956, and the top 10
factors that influence it were identified for a particular soil (USDA 1956). Although $T$ value determination criteria
have often been modified, soil formation rate remains a typical and necessary factor. Early researchers (Smith 1941;
Hays and Clark 1941; Browning et al., 1947; Klingebiel 1961) generated empirical proofs to compute this value. In
the 1980s, Pierce et al (1983, 1984a) suggested the use of a soil productivity model to calculate $T$ value and initiated
the quantitative study of this factor. Worldwide $T$ values obtained based on the soil productivity method range from
116 t/(km$^2$·a) to 9300 t/(km$^2$·a) depending on location (Pierce et al., 1983, 1984a, 1984b; Skidmore et al., 1982). In
India, the default soil loss tolerance limit of 11.2 Mg ha$^{-1}$·yr$^{-1}$ is followed to project soil conservation activities.
Scholars who examined related topics opined that criteria should be developed to determine $T$ value limits and that
these values should differ for each soil series (Pretorius 1989). Stamey and Smith (1964) proposed a notion model
of an estimated $T$ value in relation to the strength of both soil properties and soil formation rates. Skidmore (1982)
improved the concept model and calculated this value with soil thickness instead of soil characteristics. Both high
and low $T$ limits are incorporated in this approach. According to Bazzoffi (2009), the notion of tolerance erosion
based on only soil productivity and soil reformation rate is declining, and the off-site effects of soil erosion should
be considered. Therefore, this researcher suggested expanding the concept of hydrogeological risk to soil erosion by
implementing the notion of $T$ alongside a new concept, namely, environment risk of soil erosion. Scholars agree that
soil loss should stabilize soil fertility and long-term soil productivity in addition to maintaining the balance between
soil loss rate and soil formation rate (Schertz 1983; Pierce et al., 1984; Alexander 1988a, b). Lithologic soil, such as
the purple soils (entisols) derived from limestone bedrock in China, have a faster formation rate than other soils.
Under exposed conditions, the maximum weathering rate of this soil type is 15,000 Mg km$^{-2}$ yr$^{-1}$ (Zhu et al., 1999).
Purple soils are ideal for $T$ research conducted over a short time scale given their high formation rate. Thus, the
objectives of our research are to: (i) measure the soil formation rate of either the parent materials of purple soil or
the bedrock in the field (measured SR) and (ii) compare the measured and estimated SR values as well as determine
the $T$ values of purple soil. Although various influencing factors were identified when this value was first presented
in the United States in 1956 (USDA 1956), global studies on $T$ are mainly based on soil formation rate (Li et al.,

2005).

In the carbonate mountain areas of South China, soil thickness generally ranges from 30 cm to 50 cm. Once soil



is lost, the underlying basement rock is exposed, and karst rocky desertification land appears (Wang et al., 2004).
This occurrence, which is caused by soil erosion, is among the most serious eco-environmental problems in this
region. Mineralogical and geochemical studies indicate that soil layers are predominantly derived from residues
(argillaceous material) that remain after the dissolution of the underlying carbonate rocks and of the thin
argillaceous layers interbedded among these rocks (Wang et al., 1999). Owing to the low concentrations of
acid-insoluble components, the volume of carbonate rocks tends to decrease sharply in association with the
formation of weathering crusts. Highly pure carbonate rocks correspond to low acid-insoluble substance content;
therefore, the weathering–pedogenesis of carbonate rocks is the most fundamental and common
geological–geochemical process (Liu et al., 2009). This process is also the main soil formation method used in
subtropical carbonate regions. The severity of soil erosion depends strongly on the soil formation rate in the
background conditions of the geological environment. Therefore, the $T$ in carbonate areas can be determined
according to this rate.

The objectives of this research were to: (1) Discover the spatial heterogeneity and diversity of soil erosion

tolerance in the carbonate areas of south China, and disprove the old "one region, one $T$ value" concept. (2)
Proposed a new viewpoint: in karst regions, a large soil erosion modulus does not correspond to severe soil erosion,
and clarified the heterogeneity of $T$ values and the effects of this value on the erosion risk in karst
eco-environments.
**2 Study area**

The study area is located across the Yangtze River and the Pearl River in southwestern China. The

approximate coordinates are 22°01′–33°16′N and 98°36′–116°05′E. The area covers Guizhou Province, Yunnan
Province, Guangxi Zhuang Autonomous Region, Hunan Province, Hubei Province, Sichuan Province, Chongqing
Municipalities, and Guangdong Province (Fig.1). Moreover, the study area belongs to the tropical moist and
subtropical moist regions, which include different types of landforms, the annual average temperature is 11.0-19.0
degree Celsius; Because of the plenty rain, more than 80% of the area's average annual total precipitation is
between 1100 and 1300 mm, the quantity of rain throughout seasons is uneven, more rainfall in May-October,
precipitation of June to August accounted for about half of the total, but light and rainfall, temperature change
basically synchronous. Carbonate rock covers outcropped area of 522,100km², from the Sinian to Triassic, the thick
carbonate formation was deposited in the study area. Yunnan, Qianxi - Qiannan, Western Guangxi is mainly thick
layer of bare limestone, dolomite and limestone; Northeast Guizhou, Chongqing, Hubei, Xiangxi trough valley area
is mainly dolomite and clastic rocks interbedded; the middle part of Hunan, central Guilin area- southeast Guangxi
and Northern Guangdong belong to covered carbonate rock; the west of Sichuan and Yunnan consist primarily of
buried limestone. The southwestern karst mountainous areas are characterized by limestone soil, and the distribution
of this soil varies considerably. Mountainous regions with world-famous karst rock formations account for 70% of
the total area. Finally, this region is under a typical subtropical monsoon moist climate and a natural karst
mountainous environment. This area also contains inland plateau lands.



**Evaluating the spatial heterogeneity of soil loss tolerance and its effects on erosion risk in the carbonate areas of South China**

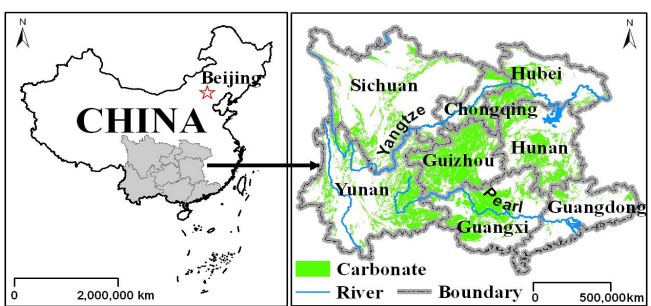

**Figure1.** Map showing the location and the distribution of carbonate regions in South China
**3 Materials and methods**
**3.1 Construction of a carbonate rock assemblage distribution map**
A 1:500,000 scale digital geological map is constructed that shows the distribution of carbonate rock assemblage
types in the carbonate areas of South China; an officially published map is used as a data source.
The method of constructing a carbonate rock assemblage distribution map is identical to our previously used
technique (Wang et al., 2004). The amount of argillaceous material in formations is considered an indicator for
distinguishing rock assemblages because this amount indicates surface soil thickness. Thus, assemblages can be
divided into three types:
(1) Homogenous carbonate rock (HC): > 90% carbonate rock, < 10% argillaceous material, and no clear clastic
interbed. On the basis of composition, HC can be categorized into three subtypes: homogenous limestone (HL),
homogenous dolomite (HD), and mixed dolomite/limestone (HDL).
(2) Carbonate rock intercalated with clastic rock (CI): 70%–90% carbonate rock, 10%–30% argillaceous material,
and a clear clastic interbed. On the basis of composition, CI can be divided into two subtypes, namely, limestone
interbedded with clastic rock (LI) and dolomite interbedded with clastic rock (DI).
(3) Carbonate/clastic rock alternations (CA): 30%–70% and 70%–30% carbonate and clastic rocks, respectively. On
the basis of composition, CA can be categorized into two subtypes, namely, limestone/clastic rock alternations (LA)
and dolomite/clastic rock alternations (DA).
The calculation of argillaceous material can be based on 5%, 20%, and 50% for HC, CI, and CA, respectively. In
addition, carbonate rock can be computed based on 95%, 80%, and 50% for HC, CI, and CA, respectively.
**3.2 Method of computing soil information rate**

The soil information rate of carbonate rocks is related to temperature, precipitation, hydrology, vegetation and

other environmental conditions. This rate changes annually, monthly, daily, and even hourly on the same day (over
daytime and nighttime). Average soil information rate can reflect overall characteristics, but it does not represent
specific position and special time. The soil information rate ranges from 30.00–89.70 mm/ka in the carbonate areas of



South China as per a long-term field observation; the mean rate is 55.27 mm/ka. As per the results of an in-house
laboratory investigation, the densities of calcite carbonatite and dolomite carbonatite are 2.75 and 2.86 t/m³,
respectively. The soil formation rate of other rock types is 200 t/(km²·a) (Li et al., 2006), and the rates of different
rock type assemblages serve as their *T* values.
Specific *T* value can be calculated with the following equation:
$$T = v \cdot Q \cdot \rho \, C + R \cdot (1-C) \qquad (1)$$
Where *T* is soil loss tolerance (t·km$^{-2}$·yr$^{-1}$); *v* is the dissolution velocity of carbonate rocks (m³·km$^{-2}$·yr$^{-1}$); *Q* is the
content of acid-insoluble components (%); $\rho$ is carbonate density (t·m$^{-3}$); *C* is the proportion of carbonate; and *R* is
the soil formation rate of other rock types.

**3.3 Construction of a KRD land distribution map in Guizhou Province in 2000**

On the basis of this classification scheme (Table 1) and in combination with the corresponding 1:100,000 scale
digital land use maps, the human–computer interactive interpreting method was used to construct a 1:100,000 scale
digital hydrogeology map, relief map, soil distribution map, and KRD land distribution maps in the year 2000 from
Landsat images.
**Table.1** The classification criterion and characteristic code of KRD types

| Classification and code of KRD type | Proportion percentage of bare rock (%) | Distribution character of the exposed rock | Color of the RS image |
|---|---|---|---|
| No KRD (NKRD) | <20 | Star | Scarlet |
| Potential KRD (PKRD) | 20-30 | Star, Line | Shocking pink |
| Already KRD (AKRD) | >31 | Patch | Pink, Gray, White |

*Note: Color of the RS image displayed with Landsat TM bands 4, 3 and 2 (displayed as red, green and blue).*

The study area measures 1,951,375 km2; therefore, much time and money must be spent for investigation. Guizhou
Province measures 176,000 km2 and lies in the center of the Southeast Asian karst zone (Fig. 2). Carbonate rock is
widespread and accounts for 62% of the total land area; in this region, karst rocky desertification is a serious
problem (Wang et al., 2004). Therefore, the relationship between karst rocky desertification and T value is
determined when Guizhou Province is taken as an example. As per this classification, a 1:100,000 scale digital map
that shows KRD land distribution overlaps with a T distribution map. The spatial relationship between these two
maps is then analyzed.

**4 Results and Discussion**

**4.1 Spatial distribution of carbonate rock assemblages**

As shown in Fig. 2a and Table 2, carbonate is mainly concentrated in Guizhou, eastern Yunan, center and western




Guangxi, western Hubei, Southeastern Chongqing, southern Hunan, northern Guangdong, and southwestern
Sichuan. The total area measures 527,196 km²; 109,416, 108,828, and 81,772 km² belong to Guizhou, Yunan, and
Guangxi, respectively. HL covers 134,996 km² and is primarily distributed in western, southern, and southwestern
Guizhou, eastern Yunan, and western Guangxi. However, this limestone is slightly scattered in Hunan. HD covers
58,723 km² and is exposed in the form of elongated belts in various places; other assemblage types are scarce. HDL
covers 63,819 km² and is mainly found in Guangxi and Hunan. Northern central and southern Guizhou. LI covers
148,577 km² and is the most widespread type of carbonate rock. DI covers 22,889 km² and is chiefly detected in
central Guizhou and southwestern Sichuan. LA covers 55,527 km² and is mainly detected in southern Guizhou and
western Hubei. Finally, DA covers only 42,665 km² and is primarily found in southwestern Sichuan and eastern
Yunan.

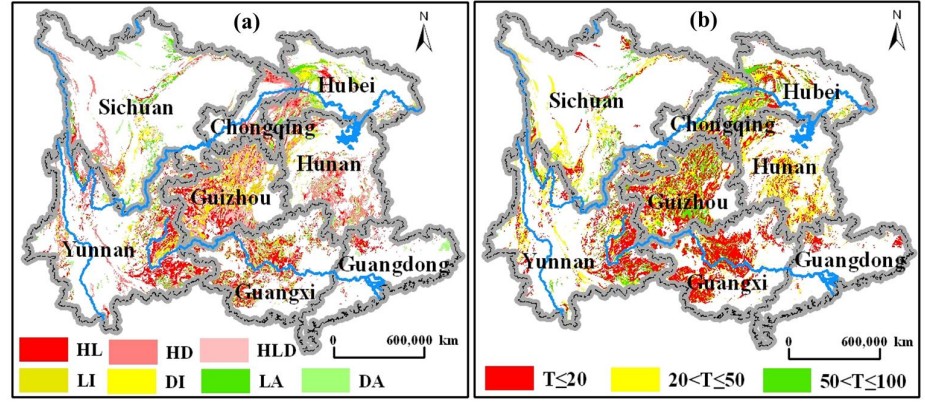

**Figure2.** Distribution map of carbonate rock assemblage types (a) and T value (b) in carbonate areas of South
China.
**Table.2** Distribution area of different carbonate rock assemblage types in carbonate areas of South China

|  | Chongqing | Guangdong | Guangxi | Guizhou | Hubei | Hunan | Sichuan | Yunan | Study area (m²) |
|---|---|---|---|---|---|---|---|---|---|
| Total | 82,400 | 179,800 | 236,300 | 176,100 | 185,900 | 21,1875 | 485,000 | 394,000 | 1,951,375 |
| Carbonate | 29,896 | 10,440 | 81,772 | 109,416 | 53,146 | 65,780 | 67,918 | 108,828 | 527,196 |
| HL | 6,722 | 4,603 | 34,309 | 30,677 | 5,184 | 9,087 | 7,579 | 36,835 | 134,996 |
| HD | 2,474 | 0 | 3,131 | 22,991 | 10,393 | 4,101 | 3,458 | 12,175 | 58,723 |
| HDL | 2,006 | 3,143 | 26,162 | 3,690 | 4,694 | 12,071 | 7,484 | 4,568 | 63,819 |
| LI | 11,114 | 2,694 | 12,355 | 19,340 | 14,641 | 35,683 | 26,085 | 26,666 | 148,577 |
| DI | 58 | 0 | 260 | 7,210 | 2,664 | 3,193 | 7,730 | 1,774 | 22,889 |
| LA | 6,835 | 0 | 5,517 | 25,231 | 6,374 | 483 | 1,889 | 9,197 | 55,527 |
| DA | 687 | 0 | 38 | 276 | 9,196 | 1,161 | 13,693 | 17,613 | 42,665 |

**4.2 Determination of T value and assessment of soil erosion risk**



Fig. 2b shows the $T$ values of different carbonate rock assemblages as calculated according to Equation (1). Those
in the HC, HL, and HDL areas are 17.22, 17.51, and 17.36 t/(km$^2$·a), respectively, whereas the $T$ values in the LI
and DI areas are 46.08 and 46.02 t/(km$^2$·a), respectively. The $T$ values in LA and DA areas are 103.80 and 107.95
t/(km$^2$·a). These values indicate the spatial heterogeneity in the carbonate areas of South China; this heterogeneity is
closely related to the amount of argillaceous material in formations that determine surface soil thickness. The "one
region, one $T$ value" concept cannot fully reflect the essence and the real circumstances in the area, and this
inadequacy may explain the diverse results obtained by different researchers. An incorrect value is typically
obtained regardless of the calculated $T$ value, and three criteria should be considered instead of only one criterion.
The $T$ values of the HC, CI, and CA areas are 20, 50, and 100 t/(km$^2$·a), respectively. These areas contain the least,
lesser, and great amounts of argillaceous materials; therefore, the three areas are each extremely, severely, and
moderately sensitive to soil erosion. Hence, the $T$ values in the carbonate areas of South China are spatially
heterogeneous. (Tab.3)
**Table.3** Criterion of T value and sensitivity of soil erosion in carbonate areas of South China

| Carbonate Rock Assemblages | T value t/(km$^2$·a) | Area (km$^2$) | Proportion (%) | Sensitivity of soil erosion |
|---|---|---|---|---|
| Homogenous carbonate rock | 20 | 257538 | 48.85% | Utmost |
| Carbonate rock intercalated with clastic rock | 50 | 171466 | 32.52% | Severe |
| Carbonate/clastic rock alternations | 100 | 98192 | 18.63% | Moderate |


In addition, the $T$ values of limestone and dolomite are similar given the same amount of argillaceous material.
According to the result of our in-house laboratory investigation, however, the dissolution velocity of calcite is 16
times that of dolomite (Drever 1997). These two types of mineral constituent rocks differ by 1.5–2 times as per both
in-house laboratory and field observations (Cao et al., 2009). In the same season and under similar spring conditions,
the carbonate content of the dolomite area in the water exceeds that of the limestone area (Jiang et al., 1997). In
terms of lithology, dolomite voidage is uniform and dense, such that the specific surface area of water–rock
interaction can be increased. As a result, conditions are set for water retention and interaction time extension (Cao et
al., 2009). Dolomite weathering is extremely intense and induces the loosening and easy formation of storage
cataclasites given the uniformity of this process. This occurrence establishes conditions for plant growth. Biological
processes accelerate dissolution velocity further; in addition, dolomite releases abundant magnesium ions during the
weathering–pedogenesis of carbonate rocks as the main action in the formation of clay mineral. By contrast,
limestone cannot supply a sufficient amount of such ions. These phenomena accelerate the dissolution velocity of
dolomite and supplement the deficiency. This mechanism may explain the similarity in the $T$ values of limestone
and dolomite.
**4.3 Effect of T value on karst rocky desertification**



As illustrated in Tab.4, the AKRD land measured 18,491, 10,955, and 9,456 km$^2$ in the extremely, severely, and
moderately sensitive areas, respectively. KRD land is concentrated in the extremely sensitive area ($T = 20$) and
covers over 47% of the total area in Guizhou Province. Of the total AKRD land, 28.16% is in severely sensitive ($T =$
50), and 24.31% is moderately sensitive ($T = 100$).
**Table.4** Karst Rocky desertification area under different sensitivity

|                     | AKRD (km$^2$) | PKRD (km$^2$) | NKRD (km$^2$) |
| ------------------- | ------------- | ------------- | ------------- |
| **Moderate sensitivity** | 9,457    | 7,889         | 8,169         |
| **Severe sensitivity**   | 10,955   | 6,004         | 9,599         |
| **Utmost sensitivity**   | 18,491   | 17,926        | 20,957        |

*Note: AKRD means already karst rocky desertification, PKRD means potential karst rocky desertification, NKRD*
*means no karst rocky desertification*

These findings suggest that a low $T$ value corresponds to a large KRD land. The KRD land area is coherent in
relation to the $T$ value criterion. Nonetheless, the relationship between NKRD land and $T$ value is unchanged. Based
on the information provided in the paragraphs above, the areas of background value in different $T$ value regions ($T =$
20, 50, 100) were 57,375, 26,558, and 25,515 km$^2$.   The distribution area of KRD land is strongly affected by the
area of the background regions. Therefore, AKRD land area may not reflect the appearance of this land in different
regions, although this area indicates the distribution situation.

Tab.5 exhibits the generation of KRD land relative to different regions that are sensitive to soil erosion. This
occurrence is maximized at 41.25%, 37.06%, and 32.23% in the severely, moderately, and extremely sensitive areas,
respectively. This finding proves that the occurrence of AKRD land is unrelated to $T$ value. In other words, this
value is not the real factor that determines the KRD appearance in carbonate areas; thus, $T$ value cannot reflect soil
erosion risk although it reflects the sensitivity of soil erosion.
**Table.5** Karst Rocky desertification area percentage under different sensitivity

|                     | AKRD (%) | PKRD (%) | NKRD (%) |
| ------------------- | -------- | -------- | -------- |
| **Moderate sensitivity** | 37.06 | 22.61    | 32.02    |
| **Severe sensitivity**   | 41.25 | 22.61    | 36.14    |
| **Utmost sensitivity**   | 32.23 | 31.24    | 36.53    |


Erosion risk depends on the relationship between RL and $T$ value rather than on soil erosion intensity or $T$ value
itself. If RL $\gg T$, then risk is high although RL is low. Conversely, if RL $\ll T$, then the soil is safe although RL is
high (Tab.6)





**Table.6** Criterion for risk assessment of soil erosion in carbonate areas of South China

| Types | Range | RL /T value | Erosion risk grade |
|---|---|---|---|
| Safe | Above-critical | R>2 | Utmost safe |
| | | 1.5<R≤2 | Severe safe |
| | | 1<R≤1.5 | Moderate safe |
| **Intermediate** | **Equal** | **R=1** | **Critical point** |
| Danger | Below-critical | 0.5≤R<1 | Utmost danger |
| | | 0.2≤R<0.5 | Severe danger |
| | | R<0.2 | Moderate danger |


The occurrence of KRD land is highest in the severely sensitive area (41.25%). This result indicates that RL is
considerably greater than the *T* value and that the situation is extremely dangerous. However, these values do not
necessarily imply that RL remains considerably smaller than *T* value in the moderately and extremely sensitive
areas. Conversely, the occurrences of KRD land are 37.06% and 32.23% in these areas; such values clearly indicate
a high degree of soil erosion. Thus, the severely sensitive area is the most hazardous area.
**4.4 T value criteria in different countries**
To develop a scientific and reasonable *T* value standard, scientists in certain countries refer to adequate research
and learn from one another. Subsequently, these researchers propose *T* values with reference to the
different conditions of their respective countries. The United States Department of Agriculture Soil Conservation
Bureau established a systematic *T* value system in 1973, and the values herein range between 220 and 1120
t/(km$^2$·a). This standard is still being used at present. Several countries in Africa reported sand and clay *T* values of
150 and 180t/(km$^2$·a), respectively. The Soviet Union presented a *T* value range of 340–1090 t/(km$^2$·a), whereas
India put forward a range of 450–1120 t/(km$^2$·a). In China, *T* values of 1000, 200, and 500 t/(km$^2$·a) are reported for
the Loess Plateau, the phaeozem region of northeast China and the northern Rocky Mountain, and the hilly red soil
region of southern China and the southwest Rocky Mountain, respectively. In this work, the *T* values in the HC, CI,
and CA areas are 20, 50, and 100 t/(km$^2$·a), respectively.
Some senior scholars and scientists have conducted preliminary studies on soil erosion in the countries. Duan X.W
modified soil productivity index model was established to calculated a quantitative *T* for different black soil species
in the black soil region of Northeast China. The *T* values of the 21 black soil species in the study area ranged from 68
to 358 t/ (km$^2$·a), with an average of 141t/ (km$^2$·a). This average *T* value is 29.5% less than the current national
standard. The T value of the three different soil subgroups in the study area were: albic black soil, 106 t/ (km$^2$·a);
typical black soil, 129 t/ (km$^2$·a); and meadow black soil, 184 t/ (km$^2$·a). (Duan et al., 2012); Shui J.G based on the
view of soil nutrient balance and test data, it was suggested that soil loss tolerance in Q$_2$ red clay derived red-earth
should be lower than 300 t/ (km$^2$·a). (Shui et al., 2003); Yuan Z.K has been determined soil loss tolerance of the
purple rock hilly area in central Hunan less than 120 t/ (km$^2$·a). (Yuan et al., 2005); Chen Q.B based on theoretical
analysis, field examination and investigation, it is considered that the 200 t/ (km$^2$·a) is the rational soil loss tolerance
of sloping field in semi-arid hill-gully area of the Loess Plateau during the long period according to soil formation




velocity, top soil nutrient balance, land productivity stability in sloping field, sediment transport tolerance of the
Huanghe River course, and regional economic development and so on. (Chen et al., 2003)

262       In karst area, some scholars have done countless research in this respect, such as: Chai Z.X according to corroded
ratio and content rate of carbonate rocks count up promise amount of soil loss tolerance which is 68 t/ (km$^2$·a) in
karst area of Guangxi Autonomous Region. (Chai et al., 1989); Chen L.J through measuring the accumulated and loss
amounts of soil nutrient for the top layer soil in forest land, and analyzing the balance of N.P.K and the rate of soil
formation, the amount of soil allowed loss is approached. It is hold that, under the upper reaches of the Changjiang
River climatically condition, the upper line of soil allowed loss is 50 t/ (km$^2$·a) for developing soil from lime stone,
that is 100 t/ (km$^2$·a) for developing soil from non-carbonaceous rock. (Chen et al., 1993); Wei Q.P worked out the $T$
values of the calcareous soil area in the karst area ranged from 0.522 to 1.285 t/ (km$^2$·a), if not consider the eluviation
and normal erosion in the soil-forming process, the scope of the T value ranged from 3.24 to 8.10 t/ (km$^2$·a), but there
are some part of the argillaceous limestone, such as non-pure carbonate rocks, the soil loss tolerance could be
increased to 16.2-40.5 t/ (km$^2$·a), and believe that the upper line of soil allowed loss is 50 t/ (km$^2$·a) for karst area.
(Wei et al., 1996); Li Y.B with the average weathering dissolving rate of carbonate rocks in Guizhou being
49.67mm/ka the pedogenesis rates of different petrologic assemblages in carbonate area have been calculated and
used as the value of soil loss tolerance in carbonate areas. The soil loss tolerance in homogenous carbonate rocks area
is lower than 6.84 t/ (km$^2$·a), 45.53 t/ (km$^2$·a) in carbonate rock intercalated with clastic rock areas and 103.46 t/
(km$^2$·a) in carbonate/clastic rock alternations areas.

In this study, T value was calculated using digital-distribution map of carbonate rock assemblages type, based on
pedosphere system theory, results indicated spatial heterogeneity and diversity in such values. T value is
proportionate to the amount of argillaceous material in formations that determine surface soil thickness in
homogenous carbonate rock areas; such values are 20 and 50 t/(km$^2$·a) in carbonate rock intercalated with clastic rock
areas and 100 t/(km$^2$·a) in carbonate/clastic rock alternation areas. In fact, erosion risk is strongly dependent on the
relationship between real soil loss (RL) and T value rather than on either erosion intensity or the T value itself. These
findings may clarify $T$ value heterogeneity and its effect on erosion risk in a karst eco-environment; hence, innovative
technological assessment solutions need not be invented. Overall, this paper presents a method that provides
experience and data for reference on the related research of soil erosion of karst landform areas of international
counterparts. However, the deficiency of this article is: This study can't fully consider dry and wet deposition in
atmosphere and the contribution of acid rain to soil forming rate, it may cause a certain impact to the accuracy.
**5 Conclusions**

This study may clarify the heterogeneity of $T$ values and its effects on erosion risk in a karst eco-environment as
an alternative to inventing innovative technological assessment solutions. Our main findings are listed as follows:

(1) $T$ values are spatially heterogeneous, and a minimum of three criteria should be considered instead of only
one when investigating the carbonate areas of South China. Apparently, the "one region, one $T$ value" concept may
not apply to this region.

(2) $T$ value is proportionate to the amount of argillaceous material in formations that determine surface soil
thickness. The $T$ values in the HC, CI, and CA areas are 20, 50, and 100 t/(km$^2$·a), respectively. These three areas
are extremely, severely, and moderately sensitive to soil erosion.

(3) The generation of KRD land is unrelated to T value, although this value reflects erosion sensitivity. Erosion



risk depends strongly on the relationship between RL and $T$ value instead of on erosion intensity or the $T$ value itself.
If RL $\gg$ $T$, then risk is high despite the low RL. On the contrary, if RL $\ll$ $T$, then the soil is safe despite the high
RL.
Overall, firstly, we report the following discovery: $T$ values are spatially heterogeneous, and a minimum of
three criteria should be considered instead of only a single criterion in karst areas. In fact, our findings disprove the
old "one region, one $T$ value" concept. Secondly, we proposed a new viewpoint: in karst regions, a large soil erosion
modulus does not correspond to severe soil erosion. Although $T$ value can reflect soil sensitivity, this value cannot
indicate soil erosion risk. Thus, a low $T$ value indicates that the local soil is highly sensitive; however, the soil
erosion risk is not necessarily high. Therefore, this risk depends strongly on the ratio between real soil loss (RL) and
$T$ value instead of on erosion intensity or on $T$ value itself.
As the result of determination time of natural erosion and environmental background conditions are not very
clear, research object, method and consideration factors of soil loss tolerance are different. Therefore, it is necessary
to make further efforts to define and specify the connotation and research methods of the natural erosion and soil
loss tolerance, at the same time it studies the natural erosion and soil loss tolerance in different types of soil and
water loss comprehensively and systematically.

*Acknowledgements.* The authors gratefully thank for the financial support provided by the auspices of National Key
Research Program of China (No. 2016YFC0502300, 2016YFC0502102, 2014BAB03B00), National Key Research
and Development (No. 2014BAB03B02), Agricultural Science and Technology Key Project of Guizhou Province of
China (No. 2014-3039), Science and Technology Plan Projects of Guiyang Municipal Bureau of Science and
Technology of China (No. 2012-205), Science and Technology Plan of Guizhou Province of China (No. 2012-6015)

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
