# Peer review of "Evaluating the spatial heterogeneity of soil loss tolerance and its effects"

_Solid Earth, 2016_

## Referee Comment (RC1) · Anonymous Referee #1 · 31 Dec 2016

The present manuscript discover the spatial heterogeneity and diversity of soil erosion tolerance in the carbonate areas of south China, and disprove the old "one region, one T value" concept, proposed a new viewpoint: in karst regions, a large soil erosion modulus does not correspond to severe soil erosion, and clarified the heterogeneity of T values and the effects of this value on the erosion risk in karst eco-environments. The work is interesting and inspiring to the field of soil erosion in karst area. Therefore I recommend this paper to be published. And it is better if the authors consider the following mentioned remarks and further improve the manuscript before submitting the final version. (1) In order to let readers understand the study and progress of soil erosion and soil loss tolerance in foreign and domestic, author should add a bit of

references. In addition, there are some mistakes in the reference, such as line 352-253, I suggest author modify it and check the citation format for other documents carefully. (2) When researchers were evaluating the harm of soil and water loss in karst area, if only from the perspective of soil erosion modulus, there would be appear a trend that with the increasing harm of soil erosion, soil erosion modulus from low to high, and then from high to low, that is soil erosion to the degree of "soilless can flow", but in fact, the erosion modulus decreases, and at this time rocky desertification area is actually in the expanding stage. So, I suggest that the author can try to evaluate the damage of soil and water loss in karst area in two stages: Under the premise of soil coverage, using soil erosion modulus to evaluate the harm of water and soil loss; When soil erosion is serious and many bare rocks on the hillside, that is rock exposed rate is very high, you can evaluated the harm of water and soil loss by the increase of rocky desertification area. In general, I think the author should describe some ideas for the future research, it can bring some inspires to some researchers who are engaged in studying soil erosions.

---

## Author Comment (AC1) · 11 Jan 2017

We are very pleased to learn from your letter about revision for our manuscript which entitled "Evaluating the spatial heterogeneity of soil loss tolerance and its effects on erosion risk in the carbonate areas of South China". We greatly appreciate your suggestion concerning improvement to this paper, and it is our honor to get your help to improve us.Thank you for your patience and the reviewers for their valuable comments and advices. We have revised the manuscript according to the every single comment which made by the referees and the editors. Moreover, we have made some correction so that we hope meet with your approval. We are sending the revised manuscript according to the comments of the reviewers. We have marked the major changes in red in

this revised version.(See the manuscript) Thank you for your consideration. Sincerely yours, *Corresponding Author: Xiaoyong Bai P.S.

Response to editor and Reviewer Comments for se-2016-151

Comments#1: In order to let readers understand the study and progress of soil erosion and soil loss tolerance in foreign and domestic, author should add a bit of references. In addition, there are some mistakes in the reference, such as line 352-253, I suggest author modify it and check the citation format for other documents carefully.

Response:Valuable suggestions! Thank you for your comments. According to your suggestion, we revised the manuscript carefully. Details are in following paragraph and MS.

We added references as follows: [1]Tian, Y., Wang, S., Bai, X., Luo, G., & Xu, Y. (2016). Trade-offs among ecosystem services in a typical karst watershed, sw china. Science of the Total Environment, 566, 1297–1308. [2]Li, Y. B., Li, Q. Y., Luo, G. J., Bai, X. Y., Wang, Y. Y., & Jie Wang, S., et al. (2016). Discussing the genesis of karst rocky desertification research based on the correlations between cropland and settlements in typical peak-cluster depressions. Solid Earth Discussions, 7(3), 741-750. [3]Luo, G. J., Jie Wang, S., Bai, X. Y., Liu, X. M., & Cheng, A. Y. (2016). Delineating small karst watersheds based on digital elevation model and eco-hydrogeological principles. Solid Earth Discussions, 7, 1-28. [4]Bai, X. Y., Wang, S. J., & Xiong, K. N. (2013). Assessing spatial-temporal evolution processes of karst rocky desertification land: indications for restoration strategies. Land Degradation & Development, 24(1), 47–56. [5]Bai, X., Zhang, X., Long, Y., Liu, X., & Zhang, S. (2013). Use of 137 cs and 210 pb ex, measurements on deposits in a karst depression to study the erosional response of a small karst catchment in southwest china to land-use change. Hydrological Processes, 27(6), 822–829.

Revised of the line 352-353: Li,Y.B., Wang,S.J., Wei,C.F., & Long,J. (2006). The spatial distribution of soil loss tolerance in carbonate area in guizhou province. Earth &

Environment, 34(4), 36-40. (line 357-358)

Comment #2 When researchers were evaluating the harm of soil and water loss in karst area, if only from the perspective of soil erosion modulus, there would be appear a trend that with the increasing harm of soil erosion, soil erosion modulus from low to high, and then from high to low, that is soil erosion to the degree of "soilless can flow", but in fact, the erosion modulus decreases, and at this time rocky desertification area is actually in the expanding stage. So, I suggest that the author can try to evaluate the damage of soil and water loss in karst area in two stages: Under the premise of soil coverage, using soil erosion modulus to evaluate the harm of water and soil loss; When soil erosion is serious and many bare rocks on the hillside, that is rock exposed rate is very high, you can evaluated the harm of water and soil loss by the increase of rocky desertification area. In general, I think the author should describe some ideas for the future research, it can bring some inspires to some researchers who are engaged in studying soil erosions

Response:We greatly appreciate your valuable suggestion concerning improvement to this paper. We entirely agree with your comments! In the later work, we will adopt your suggestions that under the premise of soil coverage, using soil erosion modulus to evaluate the harm of water and soil loss,when soil erosion is serious and many bare rocks on the hillside, that is rock exposed rate is very high, evaluated the harm of water and soil loss by the increase of rocky desertification area. I think it is very helpful to improve the quality of our paper.

Please also note the supplement to this comment:
http://www.solid-earth-discuss.net/se-2016-151/se-2016-151-AC1-supplement.pdf

**Supplement:**

[revised manuscript text omitted]

---

## Referee Comment (RC2) · Anonymous Referee #2 · 16 Feb 2017

[revised manuscript text omitted]

Zhongwu Li, Chun Liu,Yuting Dong, Xiaofeng Chang, Xiaodong Ni, Lin Liu, Haibing Xiao,Yinmei Lu,Guangming Zeng. Response of soil organic carbon and nitrogen stocks to soil erosion and land use types in the Loess hilly–gully region of China. Soil and Tillage Research. 166: 1-9, 2017.

---

## Author Comment (AC2) · 10 Mar 2017

**Author's Response**

**Dear editor and reviewer,**

We are very pleased to learn from your letter about revision for our manuscript which entitled "Evaluating the spatial heterogeneity of soil loss tolerance and its effects on erosion risk in the carbonate areas of South China".

We greatly appreciate your suggestion concerning improvement to this paper, and it is our honor to get your help to improve us!Thank you for your patience and advises. We have revised the manuscript according to the every single comment which made by the editor. Moreover, we have made some correction so that we hope meet with your approval. We are sending the revised manuscript according to the comments of the reviewer. We have marked the major changes in red in this revised version.(See the manuscript)

Thank you for your consideration!

Sincerely yours,

*Corresponding Author: Xiaoyong Bai

P.S.

=====================================================================

**Response to reviewer comments for se-2016-151**

=====================================================================

Reviewer's #1: **Abstract:**

For the sentences "Results indicated spatial heterogeneity and diversity in such values; moreover, a minimum of three criteria should be considered instead of only one criterion when investigating the carbonate areas of South China given that the "one region, one T value" concept may not apply to this region." in the manuscript. The reviewer's question is "which values" and "describe the three criteria".

**Response:** Thank you for your comments! According to your suggestion, we revised the manuscript's introduction carefully. Details are in following paragraph and MS.

Results indicated a spatial heterogeneity and diversity in soil loss tolerance; moreover, a minimum of three criteria (The $T$ values in the Homogenous carbonate rock, Carbonate rock intercalated with clastic rock, and carbonate/clastic rock alternations areas are 20,50, and 100t/(km$^2$·a), respectively) should be considered instead of only one criterion when investigating the carbonate areas of South China given that the "one region, one $T$ value" concept may not apply to this region.

Reviewer's #2: **Abstract:**

For the sentences "This erosion is extreme in karst rocky desertification (KRD) land and reflects the degree of erosion risk." in the manuscript. The reviewer's question is "This is not understandble".

**Response**: Thank you for your comments! According to your suggestion, we revised the manuscript's introduction carefully. Details are in following paragraph and MS.

Karst rocky desertification was the extreme performance of soil erosion, and reflected the risk of erosion.

Reviewer's #3: **Abstract:**

For the sentences "Overall, these findings may clarify T value heterogeneity and its effect on erosion risk in a karst eco-environment; hence, innovative technological assessment solutions need not be invented." in the manuscript. The reviewer's question is "What kind of solutions"

**Response**: Thank you for your comments! According to your suggestion, we revised the manuscript's introduction carefully. Details are in following paragraph and MS.

On the basis of 1:500,000 scale digital geological map, we got a distribution map of rock assemblage type, we selected the amount of argillaceous material in formations, because that has determined the surface soil thickness. Assemblages can thus be divided into three types: homogenous carbonate rock; carbonate rock intercalated with clastic rock; and carbonate/clastic rock alternations, according to the average content of acid insoluble matter and the average weathering rate to calculate the amount of soil loss in carbonate area.

Reviewer's #4: **Introduction:**

For the sentences "Under exposed conditions, the maximum weathering rate of this soil type is 15,000 Mg km$^{-2}$ yr$^{-1}$ (Zhu et al., 1999)." in the manuscript. The reviewer's question is "Which conditions"

**Response:** Valuable suggestions! Thank you for your comments! According to your suggestion, we revised the manuscript's introduction carefully. Details are in following paragraph and MS.

When it is exposed at the surface, the maximum weathering rate of this soil type is 15,000 Mg km$^{-2}$ yr$^{-1}$ (Zhu et al., 1999).

Reviewer's #5: **Introduction:**

For the sentences "Purple soils are ideal for T research conducted over a short time scale given their high formation rate. Thus, the objectives of our research are to: (i) measure the soil formation rate of either the parent materials of purple soil or the bedrock in the field (measured SR) and (ii) compare the measured and estimated SR values as well as determine the T values of purple soil." in the manuscript. The reviewer's question is " Purple soils are ideal for T research conducted over a short time scale given their high formation rate **(Make here a paragraph, show the novelty and justify your work**) ".

**Response**:We greatly appreciate your valuable suggestion concerning improvement to this paper. Details are in following paragraph and MS.

Purple soils are ideal for *T* research conducted over a short time scale given their high formation

rate. In the carbonate mountain areas of South China, soil thickness generally ranges from 30 cm to 50 cm. Once soil is lost, the underlying basement rock is exposed, and karst rocky desertification land appears (Wang et al., 2004). This occurrence, which is caused by soil erosion, is among the most serious eco-environmental problems in this region. Mineralogical and geochemical studies indicate that soil layers are predominantly derived from residues (argillaceous material) that remain after the dissolution of the underlying carbonate rocks and of the thin argillaceous layers interbedded among these rocks (Wang et al., 1999). Owing to the low concentrations of acid-insoluble components, the volume of carbonate rocks tends to decrease sharply in association with the formation of weathering crusts. Highly pure carbonate rocks correspond to low acid-insoluble substance content; therefore, the weathering–pedogenesis of carbonate rocks is the most fundamental and common geological-geochemical process (Liu et al., 2009). This process is also the main soil formation method used in subtropical carbonate regions. The severity of soil erosion depends strongly on the soil formation rate in the background conditions of the geological environment. Therefore, the $T$ in carbonate areas can be determined according to this rate. Thus, the objectives of this research were to: (1) Discover the spatial heterogeneity and diversity of soil erosion tolerance in the carbonate areas of south China, and disprove the old "one region, one T value" concept. (2) Proposed a new viewpoint: in karst regions, a large soil erosion modulus does not correspond to severe soil erosion, and clarified the heterogeneity of T values and the effects of this value on the erosion risk in karst environments.

Reviewer's #6: **Introduction:**

For the sentences "In the carbonate mountain areas of South China, soil thickness generally ranges from 30 cm to 50 cm. Once soil is lost, the underlying basement rock is exposed, and karst rocky desertification land appears (Wang et al., 2004). This occurrence, which is caused by soil erosion, is among the most serious eco-environmental problems in this region. Mineralogical and geochemical studies indicate that soil layers are predominantly derived from residues (argillaceous material) that remain after the dissolution of the underlying carbonate rocks and of the thin argillaceous layers interbedded among these rocks (Wang et al., 1999). Owing to the low concentrations of acid-insoluble components, the volume of carbonate rocks tends to decrease sharply in association with the formation of weathering crusts. Highly pure carbonate rocks correspond to low acid-insoluble substance content; therefore, the weathering–pedogenesis of carbonate rocks is the most fundamental and common geological-geochemical process (Liu et al., 2009). This process is also the main soil formation method used in subtropical carbonate regions. The severity of soil erosion depends strongly on the soil formation rate in the background conditions of the geological environment. Therefore, the $T$ in carbonate areas can be determined according to this rate." The reviewer's question is " This should be resumed and placed before the justification and novelty of the work".

**Response:** We greatly appreciate your valuable suggestion concerning improvement to this paper. We have followed your advise to adjusted it. Details are in manuscript.

Reviewer's #7: **Study area:**

For the sentences "Moreover, the study area belongs to the tropical moist and subtropical moist regions, which include different types of landforms. The southwestern karst mountainous areas are characterized by limestone soil, and the distribution of this soil varies considerably. This area measures 1,951,375 km$^2$ and lies in the center of the Southeast Asian karst zone. Carbonate rocks are widespread and cover an area of 44,990,000 km$^2$. Furthermore, the geotectonic foundation is complex. The layer of each geologic period from the Late Proterozoic Sinian period to the Paleozoic and Cenozoic Tertiary period is distributed across different areas, and the carbonate rocks are of various thicknesses. Mountainous regions with world-famous karst rock formations account for 70% of the total area. Finally, this region is under a typical subtropical monsoon moist climate and a natural karst mountainous environment. This area also contains inland plateau lands." in the manuscript. The reviewer's question is "Please reorganize this in i) Geological setting, climate, vegetation and soil type. ".

**Response:** We greatly appreciate your valuable suggestion concerning improvement to this paper. We have followed your advise to adjusted it. Details are in following paragraph and MS.

Moreover, the study area belongs to the tropical moist and subtropical moist regions, which include different types of landforms, the annual average temperature is 11.0-19.0 degree Celsius; Because of the plenty rain, more than 80% of the area's average annual total precipitation is between 1100 and 1300 mm, the quantity of rain throughout seasons is uneven, more rainfall in May-October, precipitation of June to August accounted for about half of the total, but light and rainfall, temperature change basically synchronous. The vegetation forms is various, both subtropical evergreen broad-leaved forest vegetation zone and the valley monsoon forest of near tropical nature, mountain rain forest; both cold temperate and subalpine coniferous forest, and warm coniferous forest of the same place; both secondary deciduous broad-leaved forest, the precious deciduous forest of the distribution is extremely limited. The spatial distribution of vegetation has shown a clear transition. Carbonate rock covers outcropped area of 522,100km$^2$, from the Sinian to Triassic, the thick carbonate formation was deposited in the study area. Yunnan, Qianxi - Qiannan, Western Guangxi is mainly thick layer of bare limestone, dolomite and limestone; Northeast Guizhou, Chongqing, Hubei, Xiangxi trough valley area is mainly dolomite and clastic rocks interbedded; the middle part of Hunan, central Guilin area- southeast Guangxi and Northern Guangdong belong to covered carbonate rock; the west of Sichuan and Yunnan consist primarily of buried limestone. The southwestern karst mountainous areas are characterized by limestone soil, and the distribution of this soil varies considerably. Mountainous regions with world-famous karst rock formations account for 70% of the total area. Finally, this region is under a typical subtropical monsoon moist climate and a natural karst mountainous environment. This area also contains inland plateau lands.

Reviewer's #8: **Materials and methods**

For the sentences "A 1:500,000 scale digital geological map is constructed that shows the distribution of carbonate rock assemblage types in the carbonate areas of South China; an officially published map is used as a data source." in the manuscript. The reviewer's question is "Was constructed or was available? Please show the source of the map ".

**Response:** Thank you for your comments! According to your questions, we explained the questions, details are in following paragraph.

Types of carbonate rock assemblages are discussed using a 1:500000 scale digital-distribution map.

Reviewer's #9: **Materials and methods**

For the sentences " The method of constructing a carbonate rock assemblage distribution map is identical to our previously used technique (Wang et al., 2004)." in the manuscript, The reviewer's question is "Describe it".

**Response:** Thank you for your comments! According to your questions, we explained the questions, details are in following paragraph.

On the basis of 1:500,000 scale digital geological map, we got a distribution map of rock assemblage type, we selected the amount of argillaceous material in formations, because that has determined the surface soil thickness. Assemblages can thus be divided into three types: homogenous carbonate rock; carbonate rock intercalated with clastic rock; and carbonate/clastic rock alternations, according to the average content of acid insoluble matter and the average weathering rate to calculate the amount of soil loss in carbonate area.

Reviewer's #10: **Materials and methods**

For the sentences " (1) Homogenous carbonate rock (HC): > 90% carbonate rock, < 10% argillaceous material, and no clear clastic interbed. On the basis of composition, HC can be categorized into three subtypes: homogenous limestone (HL), homogenous dolomite (HD), and mixed dolomite/limestone (HDL). (2) Carbonate rock intercalated with clastic rock (CI): 70%–90% carbonate rock, 10%–30% argillaceous material, and a clear clastic interbed. On the basis of composition, CI can be divided into two subtypes, namely, limestone interbedded with clastic rock (LI) and dolomite interbedded with clastic rock (DI). (3) Carbonate/clastic rock alternations (CA): 30%–70% and 70%–30% carbonate and clastic rocks, respectively. On the basis of composition, CA can be categorized into two subtypes, namely, limestone/clastic rock alternations (LA) and dolomite/clastic rock alternations (DA)" in the manuscript. The reviewer's question is "Show this information in a table".

**Response:** We greatly appreciate your valuable suggestion concerning improvement to this paper. We have followed your advise to adjusted it. Details are in following paragraph and MS.

**Table. 1** Division of rock type assemblage

| Carbonate rocks | *Continuity carbonate rocks assemblage*
*(Homogenous carbonate rock＞90%)* | | |
| --- | --- | --- | --- |
| | homogenous limestone | homogenous dolomite | mixed dolomite/limestone |
| | *Carbonate rock intercalated with clastic rock*
*(carbonate rock:70%-90%)* | | |
| | limestone interbedded with clastic rock | | dolomite interbedded with clastic rock |
| | *Carbonate/clastic rock alternations*
*(carbonate rock:30%-70%)* | | |
| | limestone/clastic rock alternations | | dolomite/clastic rock alternations |
| **Clastic rocks** | Siliceous rock,metamorphic rock,magmatic rock | | |

Reviewer's #11: **Materials and methods**

For the sentences "The soil information rate of carbonate rocks is related to temperature, precipitation, hydrology, vegetation and other environmental conditions. This rate changes annually, monthly, daily, and even hourly on the same day (over daytime and nighttime). " in the manuscript. The reviewer's question is "This needs a reference".

**Response**:Thank you for your patience and careful work. We are grateful to the reviewer for pointing out this error, we have made correction according to the reviewer's comments. Details are in following paragraph and MS.

The soil information rate of carbonate rocks is related to temperature, precipitation, hydrology, vegetation and other environmental conditions  This rate changes annually, monthly, daily, and even hourly on the same day (over daytime and nighttime)(Pak, T et al., 2016; Flügel, E. 2004).

Reviewer's #12: **Materials and methods**

For the sentences "On the basis of this classification scheme (Table 1) and in combination with the corresponding 1:100,000 scale digital land use maps, the human–computer interactive interpreting method was used to construct a 1:100,000 scale digital hydrogeology map, relief map, soil distribution map, and KRD land distribution maps in the year 2000 from Landsat images." in the manuscript. The reviewer's question is "Do you mean land use maps?".

**Response:** We greatly appreciate your valuable suggestion concerning improvement to this

paper. We have followed your advise to adjusted it. Details are in following paragraph and MS.

On the basis of this classification scheme (Table 1) and in combination with the corresponding 1:100,000 scale digital land use maps, the human–computer interactive interpreting method was used to construct a 1:100,000 scale digital hydrogeology map, relief map, soil distribution map, and karst rock desertification (KRD) land use maps in the year 2000 from Landsat images.

Reviewer's #13: **Materials and methods**

For the sentences " Table.1 The classification criterion and characteristic code of KRD types" in the manuscript. The reviewer's question is "Where this criteria of classification comes from?".

**Response:** Thank you for your patience and careful work. We have followed your advise to adjusted it. Details are in following paragraph and MS.

Table.1 The classification criterion and characteristic code of KRD types (Hu et al., 2008).
Reference: Hu Yecui, Liu Yansui, Wu Peilin, et al. Rocky desertificationin Guangxi karst mounminous area: Its tendency, formation causes and rehabilitation. Transactions of the CSAE, 24(6): 96-101, 2008.

Reviewer's #14: **Results and Discussion**

For the sentences "As shown in Fig. 4a and Table 2, carbonate is mainly concentrated in Guizhou, eastern Yunan, center and western Guangxi, western Hubei, Southeastern Chongqing, southern Hunan, northern Guangdong, and southwestern Sichuan. The total area measures 527,196 km$^2$; 109,416, 108,828, and 81,772 km$^2$ belong to Guizhou, Yunan, and Guangxi, respectively." in the manuscript. The reviewer's advise is "Move this to the beginning of the paragraph".

**Response:** Thank you for your patience and careful work. We have followed your advise to adjusted it. Details are in following paragraph and MS.

As shown in Fig. 2a and Table 2, the total area measures 527,196 km$^2$; 109,416, 108,828, and 81,772 km$^2$ belong to Guizhou, Yunan, and Guangxi, respectively. Carbonate is mainly concentrated in Guizhou, eastern Yunan, center and western Guangxi, western Hubei, Southeastern Chongqing, southern Hunan, northern Guangdong, and southwestern Sichuan.

Reviewer's #15: **Results and Discussion**

For the sentences "Table.2 Distribution area of different carbonate rock assemblage types in carbonate areas of South China" in the manuscript. The reviewer's question is "Is the data presented in %? "

**Response:** Thank you for your comments! Details are in following paragraph and MS.

**Table.2** Distribution area of different carbonate rock assemblage types in carbonate areas of South China (km$^2$)

| | Chongqing | Guangdong | Guangxi | Guizhou | Hubei | Hunan | Sichuan | Yunan | Study area (m²) |
|---|---|---|---|---|---|---|---|---|---|
| Total | 82,400 | 179,800 | 236,300 | 176,100 | 185,900 | 21,1875 | 485,000 | 394,000 | 1,951,375 |
| Carbonate | 29,896 | 10,440 | 81,772 | 109,416 | 53,146 | 65,780 | 67,918 | 108,828 | 527,196 |
| HL | 6,722 | 4,603 | 34,309 | 30,677 | 5,184 | 9,087 | 7,579 | 36,835 | 134,996 |
| HD | 2,474 | 0 | 3,131 | 22,991 | 10,393 | 4,101 | 3,458 | 12,175 | 58,723 |
| HDL | 2,006 | 3,143 | 26,162 | 3,690 | 4,694 | 12,071 | 7,484 | 4,568 | 63,819 |
| LI | 11,114 | 2,694 | 12,355 | 19,340 | 14,641 | 35,683 | 26,085 | 26,666 | 148,577 |
| DI | 58 | 0 | 260 | 7,210 | 2,664 | 3,193 | 7,730 | 1,774 | 22,889 |
| LA | 6,835 | 0 | 5,517 | 25,231 | 6,374 | 483 | 1,889 | 9,197 | 55,527 |
| DA | 687 | 0 | 38 | 276 | 9,196 | 1,161 | 13,693 | 17,613 | 42,665 |

Reviewer's #16: **Results and Discussion**

For the sentences "These values indicate the spatial heterogeneity in the carbonate areas of South China; this heterogeneity is closely related to the amount of argillaceous material in formations that determine surface soil thickness. The "one region, one $T$ value" concept cannot fully reflect the essence and the real circumstances in the area, and this inadequacy may explain the diverse results obtained by different researchers." in the manuscript. The reviewer's question is "Explain this better and ad the studies of the researchers that did not identify this."

**Response:** We greatly appreciate your valuable suggestion concerning improvement to this paper. We will explain the problem. Details are in following paragraph.

Because the spatial heterogeneity in the carbonate areas of South China; this heterogeneity is closely related to the amount of argillaceous material in formations that determine surface soil thickness.

Reviewer's #17: **Results and Discussion**

For the sentences "Hence, the $T$ values in the carbonate areas of South China are spatially heterogeneous." in the manuscript. The reviewer's question is "Calculate the coefficient of variation."

**Response:** We greatly appreciate your valuable suggestion concerning improvement to this paper. We will explain the problem. Details are in following paragraph.

In fact, it can be considered that the coefficient of variation is the same as the range, standard deviation and variance. The size of the data is not only affected by the dispersion of the variables, but also by the average value of the variables. Therefore, we believe that the calculation of the coefficient of variation of the T value is not a suitable method, and the correlation between the content of our research is not very strong.

Reviewer's #18: **Results and Discussion**

For the table (details are in following paragraph) in the manuscript. The reviewer's question is " Order the classification like: Low, moderate and severe. From the low risk to the highest. Do it here and elsewhere in the paper ".

**Response:** We have followed your advise to adjusted it. Details are in following paragraph and MS.

| Carbonate Rock Assemblages | T value t/(km²·a) | Area (km²) | Proportion (%) | Sensitivity of soil erosion |
|---|---|---|---|---|
| Homogenous carbonate rock | 20 | 257538 | 48.85% | Severe |
| Carbonate rock intercalated with clastic rock | 50 | 171466 | 32.52% | Moderate |
| Carbonate/clastic rock alternations | 100 | 98192 | 18.63% | Low |

Reviewer's #19: **Results and Discussion**

For the sentences "In addition, the *T* values of limestone and dolomite are similar given the same amount of argillaceous material. According to the result of our in-house laboratory investigation" in the manuscript. The reviewer's question is " Show the reference of this work ".

**Response:** Thank you for your comments! Details are in following paragraph and MS.

In addition, the *T* values of limestone and dolomite are similar given the dimilar amount of argillaceous material. According to the result of our in-house laboratory investigation (Zhang et al., 2007).

Reference: Zhang Tianfu, Cui Zhenang, Qian Yixiong, Xie Shuyun, Bao Zhengyu. Dissolution Kinetic Characteristics of Ordovician Marine Carbonate in Central Tarim Basin. Geological Science and Technology Information, 26(3): 19-25, 2007.

Reviewer's #20: **Results and Discussion**

For the sentences "Dolomite weathering is extremely intense and induces the loosening and easy formation of storage cataclasites given the uniformity of this process. This occurrence establishes conditions for plant growth. Biological processes accelerate dissolution velocity further; in addition, dolomite releases abundant magnesium ions during the weathering–pedogenesis of carbonate rocks as the main action in the formation of clay mineral. By contrast, limestone cannot supply a sufficient amount of such ions. These phenomena accelerate the dissolution velocity of dolomite and supplement the deficiency. This mechanism may explain the similarity in the *T* values of limestone and dolomite." in the manuscript. The reviewer's advise is " This needs a reference ".

**Response:** We greatly appreciate your valuable suggestion concerning improvement to this

paper. We will explain the problem. Details are in following paragraph.

Dolomite weathering is extremely intense and induces the loosening and easy formation of storage cataclasites given the uniformity of this process. This occurrence establishes conditions for plant growth. Biological processes accelerate dissolution velocity further; in addition, dolomite releases abundant magnesium ions during the weathering–pedogenesis of carbonate rocks as the main action in the formation of clay mineral. By contrast, limestone cannot supply a sufficient amount of such ions. These phenomena accelerate the dissolution velocity of dolomite and supplement the deficiency. This mechanism may explain the similarity in the $T$ values of limestone and dolomite (Feng et al., 2013).

Reference: Feng Zhigang, Ma Qiang, Li Shipeng, Wang Shijie, Huang Wei, Liu Jiang, Shi Wenge. Weathering Mechanism of Rock-Soil Interface in Weathering Profile Derived from Carbonate Rocks: Preliminary Study of Leaching Simulation in Rock Powder Layer. ACTA GEOLOGICA SINICA, 87(1):199-132, 2013.

Reviewer's #21: **Results and Discussion**

For the sentences "Nonetheless, the relationship between NKRD land and T value is unchanged. Based on the information provided in the paragraphs above, the areas of background value in different T value regions (T =20, 50, 100) were 57,375, 26,558, and 25,515 km². The distribution area of KRD land is strongly affected by the area of the background regions." in the manuscript. The reviewer's advise is "Please specify this better".

**Response:** Valuable suggestions! Thank you for your comments! According to your suggestion, we revised the manuscript's introduction carefully. Details are in following paragraph and MS.

Nonetheless, the relationship between NKRD land and $T$ value is unchanged. Based on the information provided above, the areas of background value in different $T$ value regions ($T$ = 20, 50, 100) were 57,375, 26,558, and 25,515 km². The distribution area of KRD is strongly affected by the area of the ecological geological environment background, such as the steep and broken landscape pattern, carbonate rocks are widely distributed, and karst is strongly developed, abundant rainfall, heavy rain and the drop of river is high.

Reviewer's #22: **Results and Discussion**

For the sentences "This finding proves that the occurrence of AKRD land is unrelated to $T$ value. In other words, this value is not the real factor that determines the KRD appearance in carbonate areas; thus, $T$ value cannot reflect soil erosion risk although it reflects the sensitivity of soil erosion." in the manuscript. The reviewer's advise is "Please calculate the relation between the variables. is just a speculation".

**Response:** Thank you for your comments! We greatly appreciate your valuable suggestion

concerning improvement to this paper. We will explain the problem. Details are in following paragraph.

The coefficient of variation is another statistic that measures the variability of the observed values in the data. When comparing the variability of two or more data, if the unit of measurement is the same as the average, the standard deviation can be compared directly. However, in our research, the coefficient of variation is difficult to calculate, and is not strongly correlated with the purpose of our research.

Reviewer's #23: **Results and Discussion**

For the sentences "Several countries in Africa reported sand and clay T values of 1.5 and 1.8 t·hm$^{-2}$·yr$^{-1}$, respectively. The Soviet Union presented a T value range of 3.4–10.9 t·hm$^{-2}$·yr$^{-1}$, whereas India put forward a range of 4.5–11.2 t·hm$^{-2}$·yr$^{-1}$. In China, T values of 10, 2, and 5 t·hm$^{-2}$·yr$^{-1}$ are reported for the Loess Plateau, the phaeozem region of northeast China and the northern Rocky Mountain, and the hilly red soil region of southern China and the southwest Rocky Mountain, respectively" in the manuscript. The reviewer's advise is "Which countries and add the citation".

**Response:** Valuable suggestions! Thank you for your comments! According to your suggestion, we revised the manuscript's introduction carefully. Details are in following paragraph and MS.

In Central Africa reported sand and clay $T$ values of 150 and 180t/(km$^2$·a), respectively. The Russia presented a $T$ value range of 340-1090 t/(km$^2$·a), whereas India put forward a range of 450-1120 t/(km$^2$·a). In China, $T$ values of 1000, 200, and 500 t/(km$^2$·a) are reported for the Loess Plateau, the phaeozem region of northeast China and the northern Rocky Mountain, and the hilly red soil region of southern China and the southwest Rocky Mountain, respectively. In this work, the $T$ values in the HC, CI, and CA areas are 20, 50, and 100 t/(km$^2$·a), respectively (Yang et al., 2004)

Reference: Yang Chuanqiang, Cai Qiangguo, Fan Haoming. Process of Soil Loss Tolerance Research in the Phaeozem Region of Northeast China. Research o f Soil and Water Conservation, 11(4): 66-96, 2004.

Reviewer's #24: **Results and Discussion**

For the table "Table.7 T value criteria in different countries" in the manuscript. The reviewer's advise is "Add the reference of the works in other countries".

**Response:** We greatly appreciate your valuable suggestion concerning improvement to this paper. We revised the manuscript (delete the Table 7, and by the way of the narrative) according to the comments made by the reviewer and the editor. Details are in following paragraph and MS.

Some senior scholars and scientists have conducted preliminary studies on soil erosion in the countries. Duan X.W modified soil productivity index model was established to calculated a quantitative $T$ for different black soil species in the black soil region of Northeast China. The $T$

values of the 21 black soil species in the study area ranged from 68 to 358 t/ (km²·a), with an average of 141t/ (km²·a). This average *T* value is 29.5% less than the current national standard. The T value of the three different soil subgroups in the study area were: albic black soil, 106 t/ (km²·a); typical black soil, 129 t/ (km²·a); and meadow black soil, 184 t/ (km²·a). (Duan et al., 2012); Shui J.G based on the view of soil nutrient balance and test data, it was suggested that soil loss tolerance in $Q_2$ red clay derived red-earth should be lower than 300 t/ (km²·a). (Shui et al., 2003); Yuan Z.K has been determined soil loss tolerance of the purple rock hilly area in central Hunan less than 120 t/ (km²·a). (Yuan et al., 2005); Chen Q.B based on theoretical analysis, field examination and investigation, it is considered that the 200 t/ (km²·a) is the rational soil loss tolerance of sloping field in semi-arid hill-gully area of the Loess Plateau during the long period according to soil formation velocity, top soil nutrient balance, land productivity stability in sloping field, sediment transport tolerance of the Huanghe River course, and regional economic development and so on. (Chen et al., 2003)

Reviewer's #25: **Conclusions**

For the sentences "This study may clarify the heterogeneity of T values and its effects on erosion risk in a karst eco-environment as an alternative to inventing innovative technological assessment solutions." in the manuscript. The reviewer's question is "Which ones?"

**Response:** Thank you for your comments! We greatly appreciate your valuable suggestion concerning improvement to this paper. We will explain the problem. Details are in following paragraph.

On the basis of 1:500,000 scale digital geological map, we got a distribution map of rock assemblage type, we selected the amount of argillaceous material in formations, because that has determined the surface soil thickness. Assemblages can thus be divided into three types: homogenous carbonate rock; carbonate rock intercalated with clastic rock; and carbonate/clastic rock alternations, according to the average content of acid insoluble matter and the average weathering rate to calculate the amount of soil loss in carbonate area.

*Notes:* We have revised the manuscript about the grammar and the words in the manuscript, and the authors and co-authors agree to the amendments made by the reviewer and editor through careful reading and discussion. We have marked the major chances in red in this revised version (See the manuscript). In addition, we reviewed and updated the references. We hope that we can meet the requirements of the reviewers and editors.  Once again to express our sincere gratitude!